# Scientific Landscape of Citizen Science Publications: Dynamics, Content and Presence in Social Media

**Núria Bautista-Puig [1,2,\*]** **, Daniela De Filippo [1]** , **Elba Mauleón [1]** and **Elías Sanz-Casado [1,2]**

[1] Research Institute for Higher Education and Science (INAECU), University Carlos III of Madrid, 28903 Madrid, Spain; dfilippo@bib.uc3m.es (D.D.F.); mmauleon@bib.uc3m.es (E.M.); elias@bib.uc3m.es (E.S.-C.)
[2] LEMI (Laboratory on Metric Information Studies), Department of Library and Information Sciences, University Carlos III of Madrid, 28903 Madrid, Spain
\* Correspondence: nbautist@bib.uc3m.es; Tel.: +34-916-248-468

**Abstract:** Citizen science (CS) aims primarily to create a new scientific culture able to improve upon the triple interaction between science, society, and policy in the dual pursuit of more democratic research and decision-making informed by sound evidence. It is both an aim and an enabler of open science (OS), to which it contributes by involving citizens in research and encouraging participation in the generation of new knowledge. This study analyses scientific output on CS using bibliometric techniques and Web of Science (WoS) data. Co-occurrence maps are formulated to define subject clusters as background for an analysis of the impact of each on social media. Four clusters are identified: HEALTH, BIO, GEO and PUBLIC. The profiles for the four clusters are observed to be fairly similar, although BIO and HEALTH are mentioned more frequently in blogposts and tweets and BIO and PUBLIC in Facebook and newsfeeds. The findings also show that output in the area has grown since 2010, with a larger proportion of papers (66%) mentioned in social media than reported in other studies. The percentage of open access documents (30.7%) is likewise higher than the overall mean for all areas.

**Keywords:** citizen science; open science; altmetrics

## 1. Introduction

### 1.1. Open Science: A New Way to Engage in, Disseminate, and Experience Science

For a little over a decade, the open science movement, an umbrella term covering a host of initiatives [1], has had a considerable impact on scientific activity. The origin of the principle of making scientific knowledge publicly available can be traced back to 2002 and the open access movement that culminated in the open sharing of the results of publicly funded research. The combination of technological advances, mass data production, scientific social networks, educational resources and open source has generated a different kind of science. This broader approach, known as open science, aims to place a whole suite of scientific formats and resources within reach of anyone [2]. It constitutes an ongoing effort to make the results of publicly funded scientific research more accessible (in digital format) to the scientific and business communities and society at large [3], but its goals are obviously more ambitious. Open science constitutes a cultural change in the way researchers, educators and other stakeholders create, store, share and deliver the results of their activity [4]. A systemic change in how science is conducted and disseminated, it enables others to contribute to and collaborate in the various stages of the research effort with all manner of freely accessible data, results and protocols [5]. "Inclusion", a term pivotal to the definitions of open science, is often deemed to mean a collaborative

scientific practice that favours public participation [6,7]. Open access, open data, participatory research, and citizen science are among the most prominent open science initiatives [8].

In recent years, a number of bodies have implemented strategies to promote and consolidate open science movement policies. The first was the Budapest Open Access Initiative (BOAI) that raises the scope and use of Open Access [9]. Later, Open Access (OA) was adopted as a principle in the European Union in 2012. Ever since, all the Horizon 2020 scientific literature produced with public funding must be made freely available [10]. In the wake of a pilot project requiring researchers in nine areas of knowledge to deposit their data in a repository, in 2017 the mandate was extended to all disciplines [11]. Open science initiatives also include infrastructures. Under the OpenAIRE project, all the scientific literature produced in projects with European funding can be pooled and the papers linked to their underlying data. The third stage of the project, launched in 2017, aims to build technological and social bridges and provide scientific information exchange services, irrespective of format [12]. The European Open Science Cloud (EOSC), another initiative, was created to provide researchers with easy access to research and innovation resources and other researchers' data [13]. The European Commission has also established the Open Science Monitor, defining three categories of "open": OA publications, OA data and open scholarly communications [14]. Under the European Union's framework programmes, which constitute an example of that approach, funding has been provided for a number of open science projects.

Despite the countless advantages of open science, this movement has not been spared from controversy. The quality of OA journals, for example, has been questioned, as some have given greater importance to the fees that many charge for publishing instead of checking the completeness of the results presented in the papers [15]. On the other hand, for the purposes of scientific evaluation, not all publications have the same value, given that those papers published in a personal blog or in journals that do not have peer evaluation, would not be considered as having the same "quality" as others included in prestigious journals and indexed in international databases such as Web of Science or SCOPUS [16]. Plagiarism and intellectual property problems are other aspects that are frequently mentioned as negative consequences of "openness" since the free availability of resources would be facilitating the "improper" and "unethical" use of scientific information [17,18]. At the same time, there are those who claim that in the age of open information, the plagiarism detection process is simpler [18,19]. These discussions show that the full implementation of Open Science entails great challenges to overcome political, economic and unethical academic practices [20].

Open access, one of the most widely known features of open science, is closely related to another, responsible research and innovation (RRI), in which the reference is to social participation in science and innovation. By encompassing citizen participation, open access, gender equality, scientific education, ethics and governance (https://www.rri-tools.eu/), that notion connects open science to other areas originating in conceits such as "citizen science".

## 1.2. European Science Policy "with and for Society"

Europe has defined one of its aims to lead responsible and innovative research aligned with societal needs. That calls for both, more effective working relations among scientists, citizens and policy-makers and more robust approaches to pooling scientific evidence itself, where the outcomes of research and innovation are understood and trusted by informed citizens and benefit society as a whole [21]. The interest in responsible research and innovation has been institutionalised within the framework of the Horizon 2020 programme. Furtherance of RRI has been earmarked as one of the areas of activity under Science with and for Society (SwafS), which aims to build effective cooperation between science and society, recruit new talent for science and equate scientific excellence to ethics and responsibility. Open science covers all those concerns, inasmuch as its recommended practices foster openness, integrity and replicability in research. The aim of SwafS has been both to progressively align research and innovation in Europe with citizens' expectations, and to deepen the opportunities for scientific modes of citizenship in the Member States of the European Union. Supporting informed and

engaged citizenship is therefore essential both for the upstream shaping of what R&I delivers to society and to developing citizens' capacity to participate fully in and maximise the benefits of R&I [22].

The European Commission's commitment to efforts to confront the challenges inherent in the new approach to generating inclusive knowledge is driving the participation of scientific institutions in open science. The importance of citizen science is evidenced in its inclusion as one of the eight priorities defined in the European Commission (2018) document entitled "Open Science Policy Platform Recommendations" [21].

A number of definitions of citizen science may be found in the literature, such as: an activity open to all citizens, not only for the most "privileged" [23], enabling them to participate in the entire process, from inception, the choice of lines of research and the implementation of procedures, tasks that were traditionally confined to researchers. As the literature shows, the first citizen science (CS) activity was a 1989 Audubon Society initiative in which 225 American citizens collected rain samples to test rainwater acidity [24]. The term "citizen science" was defined in the mid-nineteen-nineties by Alan Irwin as a '' form of science developed and enacted by citizens themselves' [25] (p. 11).

It is currently defined as public engagement in the scientific research process in which citizens participate actively in different ways, with their intellectual or knowledge, tools or resources. The main aim is to "co-create a scientific culture" and an exchange of understanding [26] (p. 6).

The main benefits of citizen science include: the increase in scientific literacy and the critical capacity of citizenship, the democratisation of the scientific process, the motivation of young people to pursue scientific careers, the generation of new knowledge with perspectives of innovative research, and the expansion of the abilities of researchers [22]. This new science ethos has found significant institutional backing, such as in the European Union's Framework Programmes that have funded several citizen projects. One example is the Foster actions contained in the Seventh Framework Program involving 13 institutions in eight countries. Several candidacies for the Union's most recent H2020 Programme focused on those issues. Science with and for society (SwafS) is one example of the interest in this new approach. A number of authors [27,28] have stressed the value of this type of science in which scientific knowledge goes beyond open access to information and reach social actors hitherto overlooked, thus favouring the production of new knowledge.

The fundamental change introduced by such movements is the engagement of groups such as researchers, community organisations, politicians and others in the various stages of research, actively participating in all of them, as well as in the pursuit of solutions and their sharing with the members of the community for future use.

Of course, the growing weight of citizen science entails new questions to be solved in which technical, ethical, legal, socio-economic aspects are combined...One of them is validation and integrity in research (will the data obtained by non-specialists be as valid as those obtained by professional scientists? In a way, these are new areas of research and regulation, but at the same time, there are underlying issues related to old problems. As stated by Pelacho et al. [29], the results obtained in studies carried out with or by citizens must incorporate efficient and safe methods of validation. No one wants to be subjected to insufficiently verified therapy or to have to drive vehicles designed on the basis of non-validated criteria in the event of illness. Citizen science must be present in our future, and it is essential that it be a citizen ... but it can never cease to be Science.

## 1.3. The Challenges Facing Open Science

Up to a few years ago, the dissemination of research findings in journals and in the form of patents was the consolidated model for producing and assessing knowledge. As Callon et al. [30] explain, scientometrics and bibliometrics were the primary tools for analysing scientific output and measuring its impact on the scientific community. One distinctive feature of our age is the exponential growth of information, with a proliferation of new information and communication technologies that favour greater information exchange, scientific output and mass data production. Increasing access to the internet has ushered in a change of paradigm for accessing and publishing scientific content.

The production and dissemination of scientific knowledge have been substantially impacted by the new open science context. The change of paradigm introduced with the advent of the web 2.0 has favoured communication and collaboration among academic agents and their interaction with social agents. Today many content sharing sites such as fora, blogs and social networks (including Facebook and Twitter) have not only proven very popular among the public at large, but have also seduced members of the scientific community, giving rise to the so-called "academic social web". This new space enables researchers to share and validate their projects through tools tailored to academia, such as reference management software (Mendeley, CiteULike), professional networks (ResearchGate, ScienceOpen) and digital identity applications (Web of Science's ResearcherID or ORCID) [31].

The open access to science movement and online publications and repositories including PLos ONE, ArXiv, CiteSearch, PubMed and RePEc not only supplement such tools but, together with the ceaseless growth of informal communication channels, constitute a new challenge for researchers analysing scientific activity [32]. Again, as mentioned above, the challenge for evaluation also involves considering (and overcoming) the limitations of using these tools (such as plagiarism and lack of quality).

To rise to that challenge, new models must be designed to enhance traditional bibliometric-based research by drawing from altmetric indicators denoting societal interest in a given field of science [33]. The term "altmetrics" was coined by Jason Priem in a 2010 tweet. It is defined as the creation and study of new social web-based metrics to analyse and characterise scholarship. The term is essentially associated with the academic social web and metrics are deemed to be alternative either because they use sources of data other than traditional impact indicators or because they define and calculate new, alternative indicators.

One of the main advantages of these indicators is that as they provide information both at article [34] and at author level [35,36], the impact of studies may be appraised with no regard to the quality or visibility of the publishing journal.

The interest in the use of alternative metrics has generated a series of studies that analyse, for example, the advantages and drawbacks of altmetric indicators [35–38]. Other authors address the scope of different platforms and indicators [39,40], and numerous research papers have focused on the characteristics of documents that may affect their social impact [41–43].

The relationship between traditional academic impact and social media impact has prompted growing interest and many studies have analysed the links between impact, visibility and open access to publications [44–52].

Although serious doubts have been raised about the value of altmetric indicators (particularly in connection with bias [40,41], the lack of a transparent methodology or the obsolescence of results [37]), the transfer time from the scientific community to society is obviously shorter when results are echoed in social networks, which also reach a broader and more diverse audience.

### 1.4. Background and Objectives

Research on citizen science has not focused solely on the content and development of projects involving participatory methodology. The question of output, for instance, has been addressed by a number of authors. Others, such as Bonney et al. [53], have conducted a systematic study of the activities undertaken in citizen science projects, measuring their academic impact through a series of quantitative indicators. In 2014, Comber et al. [54] published a paper containing a semantic analysis of terms used to describe citizen sensing and crowdsourced data used in scientific analyses. They analysed journal abstracts downloaded from SCOPUS that matched any of a number of terms related to crowd sourced data and citizen science. Follet and Strezov [55], in turn, used the Web of Science and SCOPUS databases to detect and analyse publications on citizen science and their use in new research projects. Kullenberg and Kasperowski [56] later developed a precise documentary search strategy to monitor the changing use of terms related to Web of Science-listed publications on citizen science and projects culminating in scientific publications. Another recent paper described a case study involving citizen

participation, with an analysis of the variations in output on the subject in SCOPUS to contextualise the findings [57].

Taking the aforementioned studies as a starting point and given the importance acquired by citizen science in open science, this paper analyses the scientific research on the subject and its visibility in social media. Scientific publications of international prestige are explored to:

- analyse the dynamics of scientific output over time through the study of references and mentions of articles of open science
- identify the key subjects addressed in scientific papers on citizen science
- study visibility in social media by identifying the subject matters addressed and their possible dissemination patterns.

In short, the aim is to broaden the scope of bibliometric studies and ascertain whether the importance of citizen participation in this new approach to conducting research is mirrored in academia and society at large. What items are addressed in citizen science-related publications? Are certain types of subjects better suited than others to citizen participation? The most original feature of the study is the inclusion of altmetric indicators to explore form- and content-related items in greater depth. That entails analysing how documents are accessed. Are citizen science publications more prone to be disseminated in open access journals? Content is also explored, along with the relationship between these two features and visibility in social media. Are the subjects of papers related to the type of medium where results are published?

The study was conducted by stages, as described in Chapter 2 below (Sources and methodology). Chapter 3 discusses the most prominent findings, which are compared to those of earlier research in Chapter 4.

## 2. Sources and Methodology

Web of Science (WoS) was the source of the citizen science publications studied. Despite its well-known linguistic and thematic biases (better coverage of publications in English and in topics related to experimental and biomedical sciences), and the under-representation of publications from non-English speaking countries [58,59], this source of information was chosen because it lists high-quality documents that can be analysed with content and visualisation tools. The publications were retrieved from the Core Collection, which includes the Science Citation Index (SCI), Social Science Citation Index (SSCI) and Arts and Humanities Citation Index (A&HCI) databases.

The altmetric indicators of social media presence were found with API by Altmetric.com, one of the key sources of such information. According to some authors, the main issue with this type of sources is the difficulties that entail identifying mentions to scientific papers, similarly to the shortcomings found when using webometric techniques [60]. Conceptually speaking, another very serious limitation detected in previous studies is related to the sources covered by Altmetric.com. This heterogeneity affects their analysis and complicates the construction of a proper interpretative framework for these indicators [40,41]. Although also subject to certain limitations, it delivers information broken down by social media.

The steps taken to reach the objectives proposed are set out below.

- **Formulation of a search strategy:** drawing from an analysis of earlier bibliometric studies, a search strategy was formulated based on frequently used terms relevant to citizen science. That strategy, introduced in an earlier paper [61], retrieves publications by searching the title field (TI) for the following terms: "crowd science", "community science", "participatory research", "participatory action research", "community-based research", "citizen research", "science shop", "public-participation", "citizen observatory" and "community engagement research". Other terms found in similar studies [56] and sought in topics (TS), title, abstract and keywords, proved to be highly pertinent to grow the number of papers retrieved. Terms defined in previous

studies [56] as synonyms to existing search concept were included, and were checked with boolean operators and query sets in the iterations performed, such as: "biodiversity monitoring", "civic science", "eBird", "locally-based monitoring", "neogeography", "participatory GIS", "participatory monitoring", "participatory science", "PPGIS", "volunteer monitoring" and "volunteered geographic information (VGI)".

Publications were retrieved irrespective of document type and date.

- **Publication retrieval and information processing:** the information on the documents on open science was exported (in *.txt format) and a relational database formulated with MySQL, in which all the records were entered.
- **Establishment of bibliometric indicators. The study focused on the following indicators:**

(1) for activity and access

- ○ yearly variation in output
- ○ output growth rate
- ○ contribution to database
- ○ output by country (absolute values and activity index)
- ○ number and percentage of documents with a Digital Object Identifier (DOI)
- ○ number and percentage of open access (OA) documents

(2) for subject specialisation

- ○ distribution of output by WoS category
- ○ identification of document clusters based on the co-occurrence of both article and author keywords, using the VosViewer tool. Regarding the clustering, VosViewer provides its own algorithm based on modularity optimization [62,63]. In addition, normalization method used was the Ling/Long modularity

After collecting all the publications on the subject (subject to no time constraints), the analysis focused on the 12 years from 2006 to 2017 when 77.68% of the output was published. That period was subsequently sub-divided into two 6-year sub-periods for comparative analysis.

- **Establishment of altmetric indicators:** information was gathered on visibility in social media based on publications' DOI which is a unique identifier indexed in Web of Science (WoS) that allows to obtain the altmetric information for each publication. A script developed by the Carlos III University of Madrid's Information Metrics Studies Laboratory (Spanish initials, LEMI) was applied to the Altmetric.com API, which delivered the following indicators for each DOI-bearing document:[1]

- ○ percentage of documents with mentions in social media:[2]
- ○ number of mentions in blogposts, Twitter, Wikipedia, Mainstream Media (MSM), videos and newsfeeds
- ○ maximum number of mentions per document and type of source
- ○ proportion of open access (OA) documents and mentions in social media
- ○ identification of mentions by document cluster (number, yearly variation and source).

---

[1] This script programmed in python uses the "requests" library to make http or https requests from Altmetric's API (http://api.altmetric.com).

[2] The study considered all the sources of social media for which Altmetric.com offers information (https://www.altmetric.com/about-our-data/our-sources/). Mentions in academic networks (Mendeley, Connotea, CiteULike, etc.) have been excluded from the count.

## 3. Results

### 3.1. Bibliometric Indicators

#### 3.1.1. Activity and Access

The search strategy deployed retrieved 5100 documents on citizen science. Although documents were found as early as 1956, the densest concentration was in recent years, with the period 2006–2017 accounting for 77.68% (3962 documents) of the total (Figure 1). The cumulative average growth rate for this type of publications was 16.14%, much greater than for the WoS database, which grew at an average rate of 5.08% in those same years. The proportion of publications on citizen science in the database doubled in the 12 years studied (Table 1).

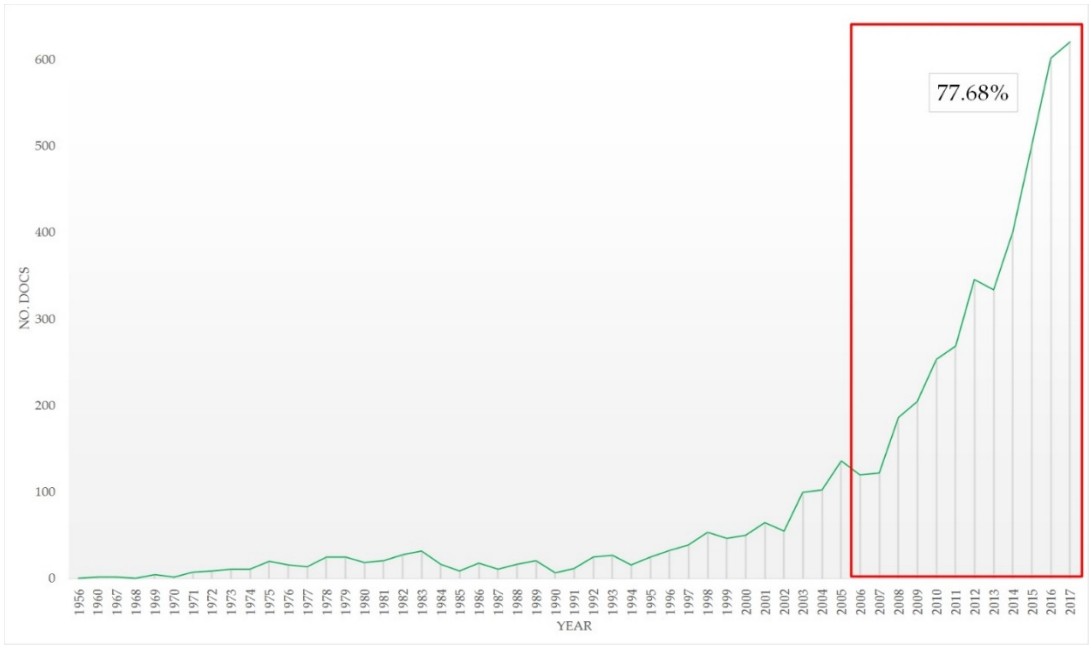

**Figure 1.** Yearly variation in the output on citizen science.

**Table 1.** Yearly variation in output on citizen science, growth rate and proportion of publications on citizen science in the Web of Science (WoS) database (2006–2017).

| Year | No. Docs on Citizen Science (CS) | No. Docs in WoS | Proportion CS Docs/WoS Docs |
|---|---|---|---|
| 2006 | 120 | 1,742,394 | 0.0069 |
| 2007 | 122 | 1,891,243 | 0.0065 |
| 2008 | 187 | 2,011,175 | 0.0093 |
| 2009 | 205 | 2,141,140 | 0.0096 |
| 2010 | 254 | 2,171,074 | 0.0117 |
| 2011 | 269 | 2,272,159 | 0.0118 |
| 2012 | 346 | 2,373,215 | 0.0146 |
| 2013 | 334 | 2,472,877 | 0.0135 |
| 2014 | 401 | 2,562,453 | 0.0156 |
| 2015 | 500 | 2,890,610 | 0.0173 |
| 2016 | 602 | 3,012,436 | 0.0200 |
| 2017 | 622 | 3,005,145 | 0.0207 |
| Total | 3962 | 28,545,921 | 0.0139 |
| Growth | 418.33 | 72.47 | |
| Cumulative average growth rate (CAGR) | 16.14 | 5.08 | |

Further to the data on national output, 171 countries produced documents on the subject, although just 51 published over 10 in the period. Output was highest in the United States with 1624 documents (41%), United Kingdom with 585 (14.77%), Canada with 463 (11.7%) and Australia with 367 (9.2%). Of the countries with output accounting for over 1% of the total, the highest activity indices for publications on citizen science were recorded for South Africa, New Zealand, Australia, Canada and Ireland (AI > 2). Output by country for both the total publications listed and the documents on citizen science (CS) is given in Table 2. On the whole, the countries with large absolute CS output also exhibited intense activity in citizen science (AI > 1), although Japan, China and India, with values much lower than expected, deviated from that pattern (Figure 2).

**Table 2.** Output by country.

| Country | Publications on Citizen Science | | Total Publications in WoS | | AI (% SC Docs/ % WoS docs) |
|---|---|---|---|---|---|
| | No. docs | % | No. docs | % | |
| USA | 1624 | 40.99 | 6,566,822 | 28.83 | 1.42 |
| England | 511 | 12.90 | 1,571,630 | 6.90 | 1.87 |
| Canada | 463 | 11.69 | 961,048 | 4.22 | 2.77 |
| Australia | 367 | 9.26 | 752,804 | 3.30 | 2.80 |
| Germany | 247 | 6.23 | 1,493,842 | 6.56 | 0.95 |
| Netherlands | 150 | 3.79 | 542,188 | 2.38 | 1.59 |
| France | 136 | 3.43 | 1,014,794 | 4.46 | 0.77 |
| Italy | 136 | 3.43 | 907,197 | 3.98 | 0.86 |
| China | 132 | 3.33 | 2,499,931 | 10.98 | 0.30 |
| Spain | 102 | 2.57 | 759,415 | 3.33 | 0.77 |
| Switzerland | 90 | 2.27 | 396,598 | 1.74 | 1.30 |
| South Africa | 88 | 2.22 | 142,965 | 0.63 | 3.54 |
| Brazil | 77 | 1.94 | 508,986 | 2.23 | 0.87 |
| Sweden | 76 | 1.92 | 339,807 | 1.49 | 1.29 |
| Scotland | 75 | 1.89 | 226,297 | 0.99 | 1.91 |
| New Zealand | 68 | 1.72 | 121,858 | 0.53 | 3.21 |
| Denmark | 67 | 1.69 | 219,693 | 0.96 | 1.75 |
| Austria | 60 | 1.51 | 208,690 | 0.92 | 1.65 |
| Finland | 60 | 1.51 | 160,014 | 0.70 | 2.16 |
| Belgium | 58 | 1.46 | 293,505 | 1.29 | 1.14 |
| Norway | 57 | 1.44 | 158,867 | 0.70 | 2.06 |
| Portugal | 56 | 1.41 | 168,637 | 0.74 | 1.91 |
| Ireland | 55 | 1.39 | 124,743 | 0.55 | 2.53 |
| Japan | 49 | 1.24 | 1160,504 | 5.09 | 0.24 |
| Mexico | 44 | 1.11 | 158,509 | 0.70 | 1.60 |
| India | 42 | 1.06 | 668,574 | 2.94 | 0.36 |

A total of 3416 (86.22%) of the publications retrieved had a DOI. The number with that identifier grew across the period analysed, from 74.17% in 2006 to 91.96% in 2017 (Figure 3).

The 1215 open access documents on citizen science represented 30.67% of the total for the 12 years, during which the percentage grew from 12.5% to 37.5% (Figure 3). That compares to the 23.43% of open access (OA) documents in the WoS database as a whole in the same period. By category, 23.54% of all the open access publications on citizen science were "green", 55.64% "gold" and 20.82% 'bronze".[3]

---

[3] Gold route is the commonly used and the author pays an article-processing charge (APC) to the publisher at publication time and the publisher makes the document available freely and accessible for everyone. Green OA is linked to the concept of self-archiving: it refers to the authors' ability to publish the results of their research in OA, archiving their work in a repository for scientific publications (institutional, non-commercial or commercial) or in their personal web page [7]. With bronze OA the document is available to read on the publishers' webpage but without a license that allows re-use of content.

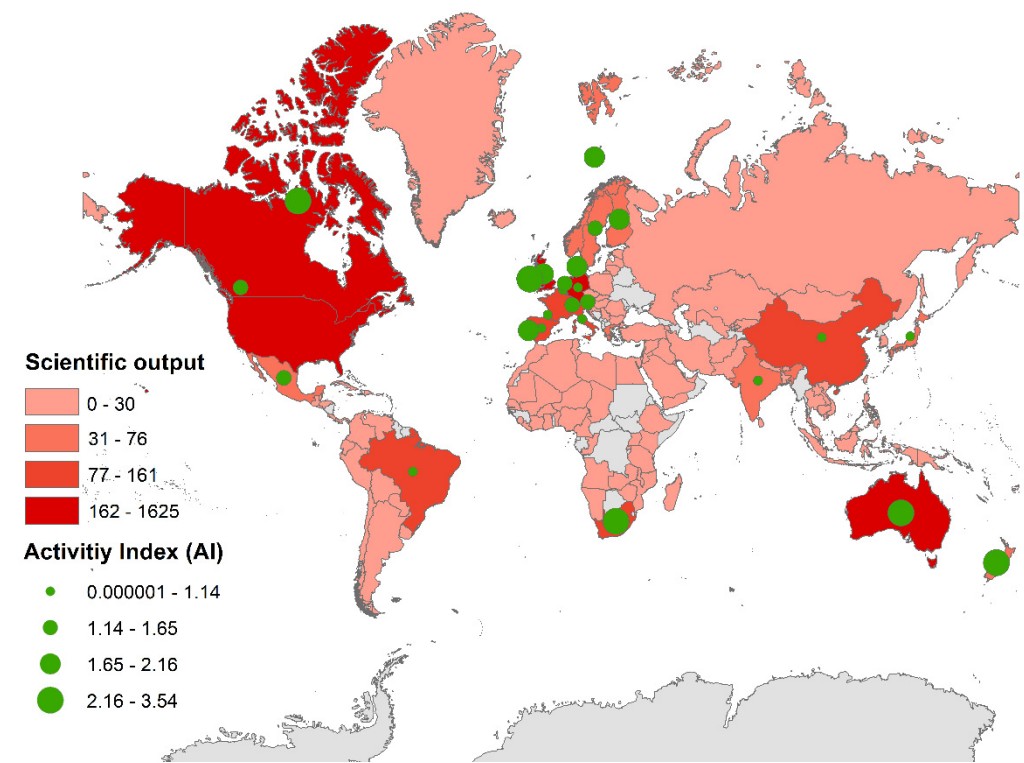

**Figure 2.** Publications on citizen science: output and activity index by country.

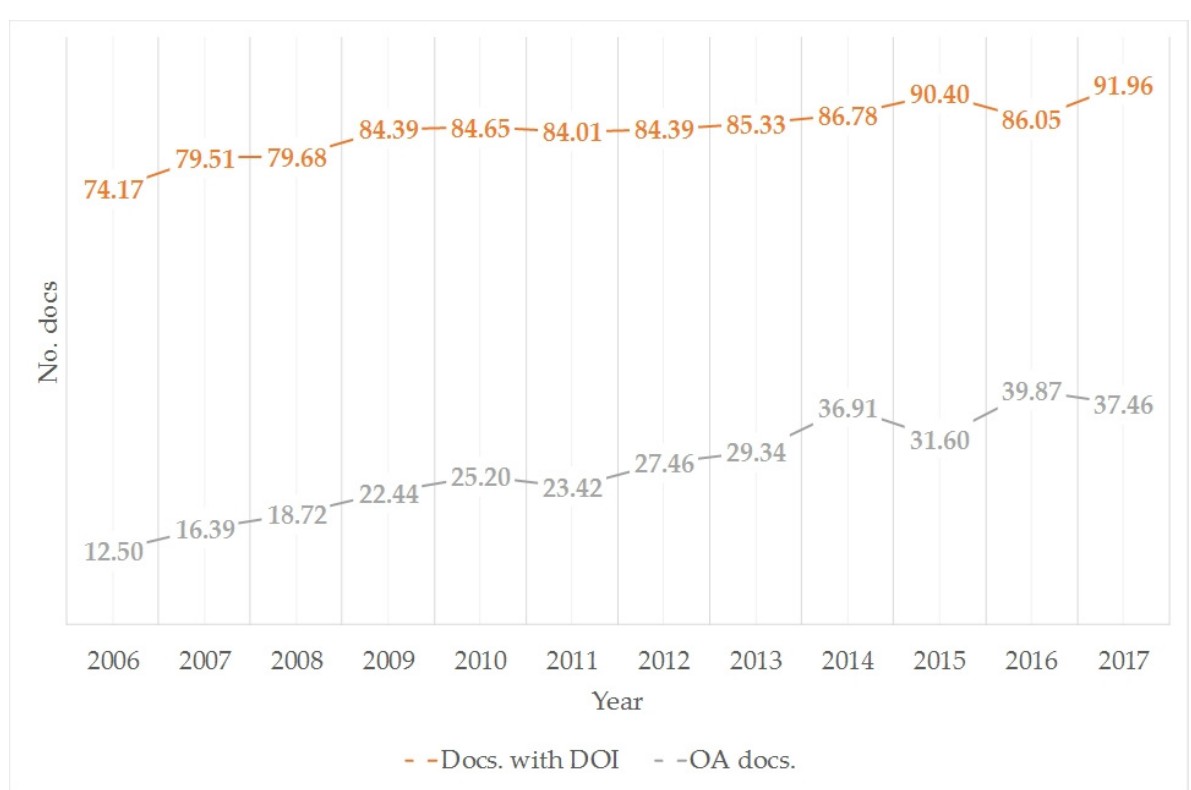

**Figure 3.** Documents with Digital Object Identifier (DOI) and open access documents, per cent of total by year.

### 3.1.2. Subject Specialisation

The WoS categories under which articles on citizen science were most frequently published across the period studied (2006–2017) were Environmental Science with 526 documents (13.27%), Environmental and Occupational Health with 524 (13.22%), Ecology with 505 (12.75%) and Geography with 380 (9.59%). Differences were observed in the top 20 categories in the two periods studied (Figure 4). Whilst in the first period, Environmental Sciences, with 143 documents, accounted for 12% of the total output on citizen science, in the second the proportion rose to 14% with 383 documents. Other subject categories for which participation also grew included Ecology (from 12% in the first to 13% in the second period) and Geography (from 8% to 10%). Others, such as Public, Environmental and Occupational Health, declined (from 17% in the first to 11% in the second period) and Social Sciences, Interdisciplinary (from 6% to 5%).

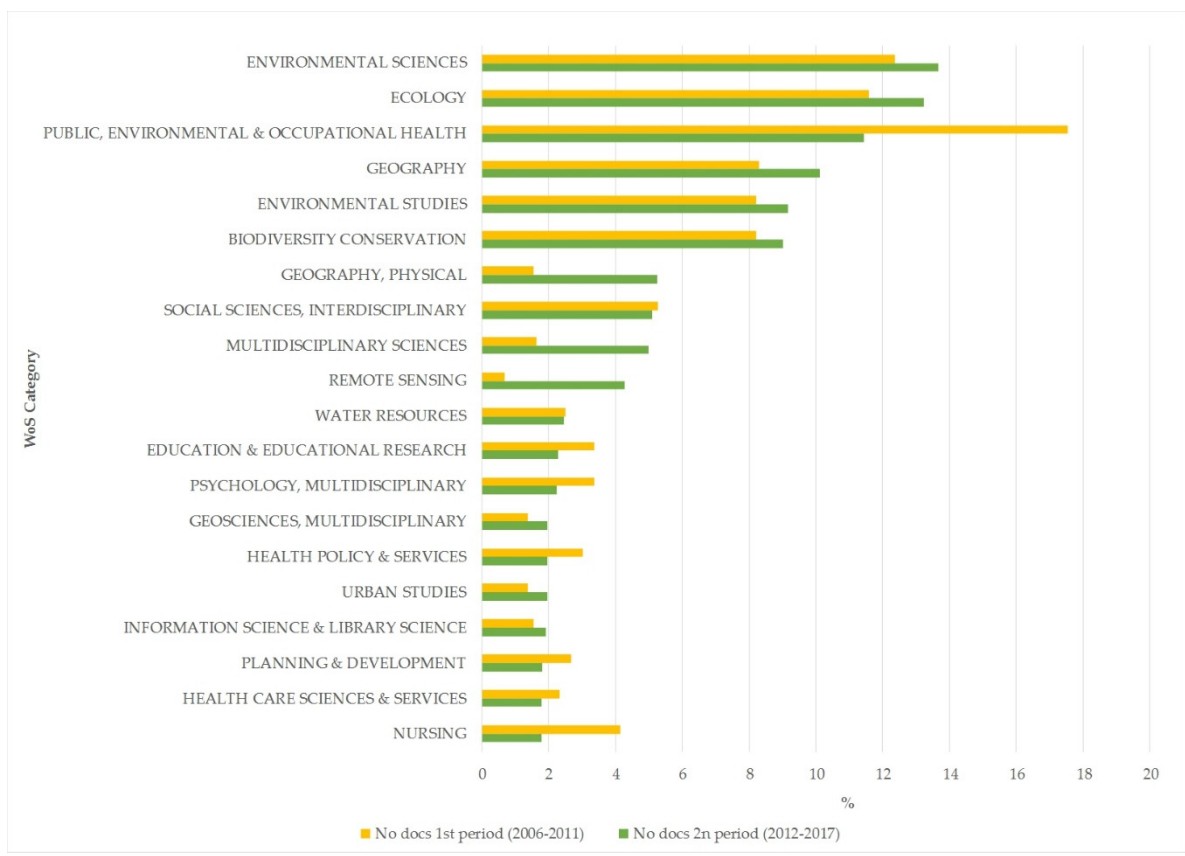

**Figure 4.** Distribution of output by WoS category (top 20 disciplines).

Keyword co-occurrence-based clustering identified four main subject groups for the overall period (2006–2017), which were defined on the grounds of the terms involved as follows: HEALTH–Participatory Research (Cluster 1), BIO–Science (Cluster 2); GEO–Participation in GIS; (Cluster 3) and PUBLIC–Citizen Participatory (Cluster 4). The number of documents in each cluster and the terms most frequently associated with them are given in Table 3 (where some documents may be classified under more than one cluster).

**Table 3.** Classification of publications by cluster.

| Cluster | No. Docs | Top 10 Words |
|---|---|---|
| Cluster 1 HEALTH–Participatory Research (red) | 1685 | community-based participatory research; health; participatory action research; participatory research; community; challenges; public health; care; united states; project |
| Cluster 2 BIO–Science (green) | 1613 | citizen science; conservation; management; biodiversity; tools; biodiversity monitoring; diversity; climate-change; ecological research; indicators |
| Cluster 3 GEO–Participation in GIS (blue) | 1170 | volunteered geographic information; GIS; OpenStreetMap; PPGIS; quality; information; systems; participatory GIS; crowdsourcing; ecosystem services |
| Cluster 4 PUBLIC–Citizen Participatory (yellow) | 1124 | public participation; science; knowledge; participation; policy; governance; framework; citizen participation; decision-making; sustainability |

The keyword co-occurrence map from which the four aforementioned clusters and the relationships among them across the entire period (2006–2017) were drawn is reproduced in Figure 5. Node size is indicative of the number of documents, whilst the lines identify inter-document relationships and their thickness, intensity. A first cluster, labelled HEALTH, included terms related to that topic and to methods of citizen participation such as "participatory action research", "community-based research" and "action research". Other health-related terms were also detected, such as "cancer", "health promotion" and "prevention". A second cluster, labelled BIO, included terms such as "conservation", "biodiversity monitoring", "ecology", "protected areas" and "indicators". Another prominent term in this cluster bore the label "citizen science". The cluster labelled BIO was connected to GEO and PUBLIC via "management". The terms appearing in the cluster labelled GEO included "volunteered geographic information", "participatory GIS" and "public participation GIS". The PUBLIC cluster contained terms such as "public participation", "governance", "policy" and "engagement".

Here also, an analysis of the two periods separately revealed differences. Three clusters were identified in the first period (2006–2011). One was HEALTH and another BIO, in which the citizen science node had not yet attained the importance observed in the second period. The third node was deemed to be the precursor of what would subsequently be the GEO and PUBLIC clusters, in light of the associated terms: "participatory GIS" and "public participation", among others.

In the second period (2012–2017), GEO and PUBLIC no longer shared the same keywords. Cluster no. 3 GEO was associated with the development of new technologies, with the appearance of terms such as "web 2.0", "neogeography", "OpenStreetMap" and "Twitter", among others. Cluster no. 4 PUBLIC, in contrast, included words such as "community", "public participation", "decision-making", "democracy", "engagement", "policy" and "science". In the second period, the terms "citizen science" and "conservation" gained significance in the BIO cluster (Figure 6).

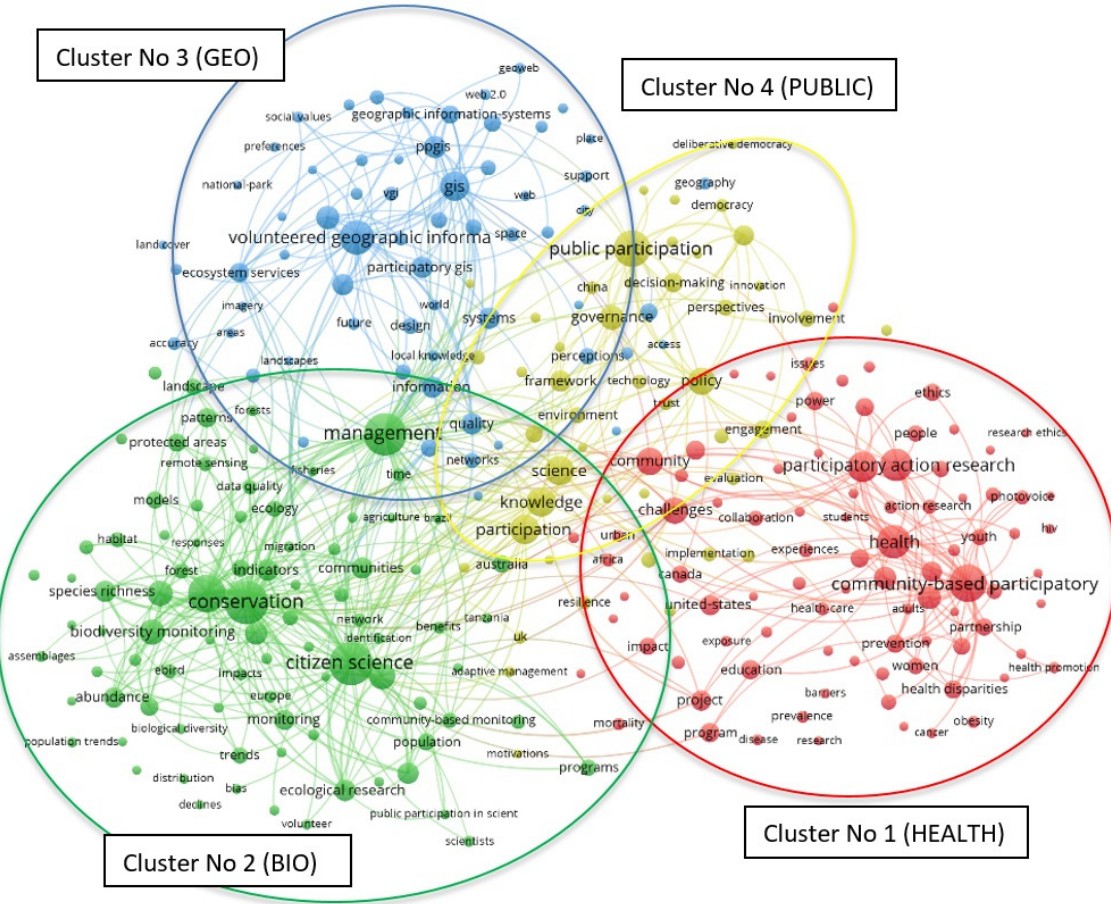

**Figure 5.** Co-occurrence map of subject areas with ≥15 documents on citizen science (CS) across the entire period.

The keywords that appeared frequently in all the clusters were "conservation", "management", "biodiversity" and "health". Certain differences were detected, however, in an analysis of the most frequent terms in each cluster and their variation across the period.

In Cluster 1, HEALTH terms such as "participatory research" grew more than methodology-related phrases such as "community-based participatory research". The number of studies on health as well as the term "challenge" also grew in this cluster.

The frequency of terms such as "conservation" and "biodiversity" grew most in Cluster 2, BIO. The frequency of others such as "biodiversity monitoring" remained unchanged throughout.

The prevalence in Cluster 3, GEO of terms such as "volunteered geographic information" and "OpenStreetMap" (a collaborative project to create editable, open source maps) grew across the period, in particular in the most recent years. That rise was related to greater interest in subjects such as "quality" and "information", the frequency of which rose in the second half of the period.

The frequency of terms such as "participation", "policy" and "knowledge" rose in Cluster 4, PUBLIC, particularly in 2012–2017.

The variations in the most prominent terms in each cluster are shown in Figure 7.

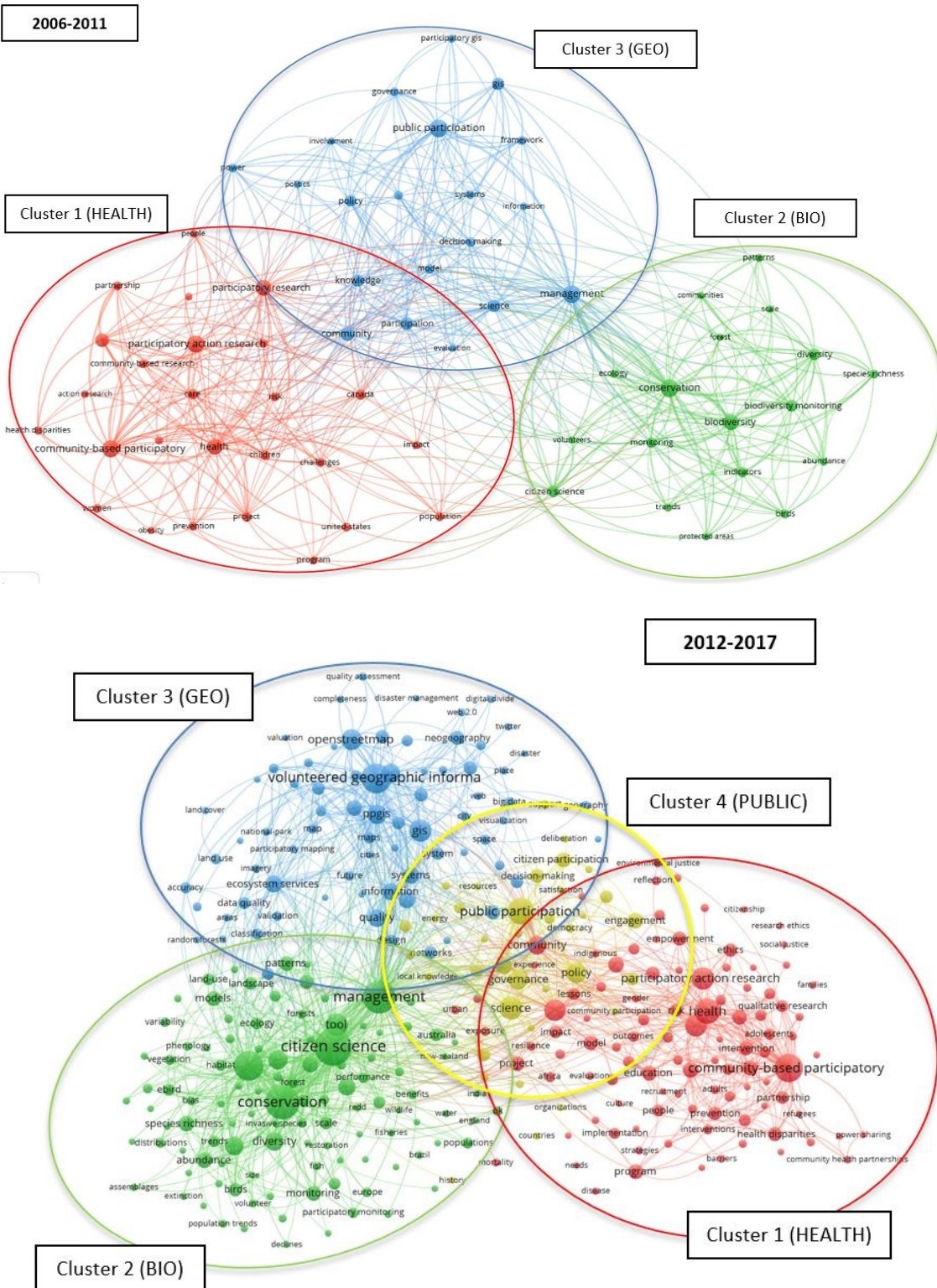

**Figure 6.** Co-occurrence map of subject areas with ≥15 documents on citizen science (CS): comparison between two 6-year periods.

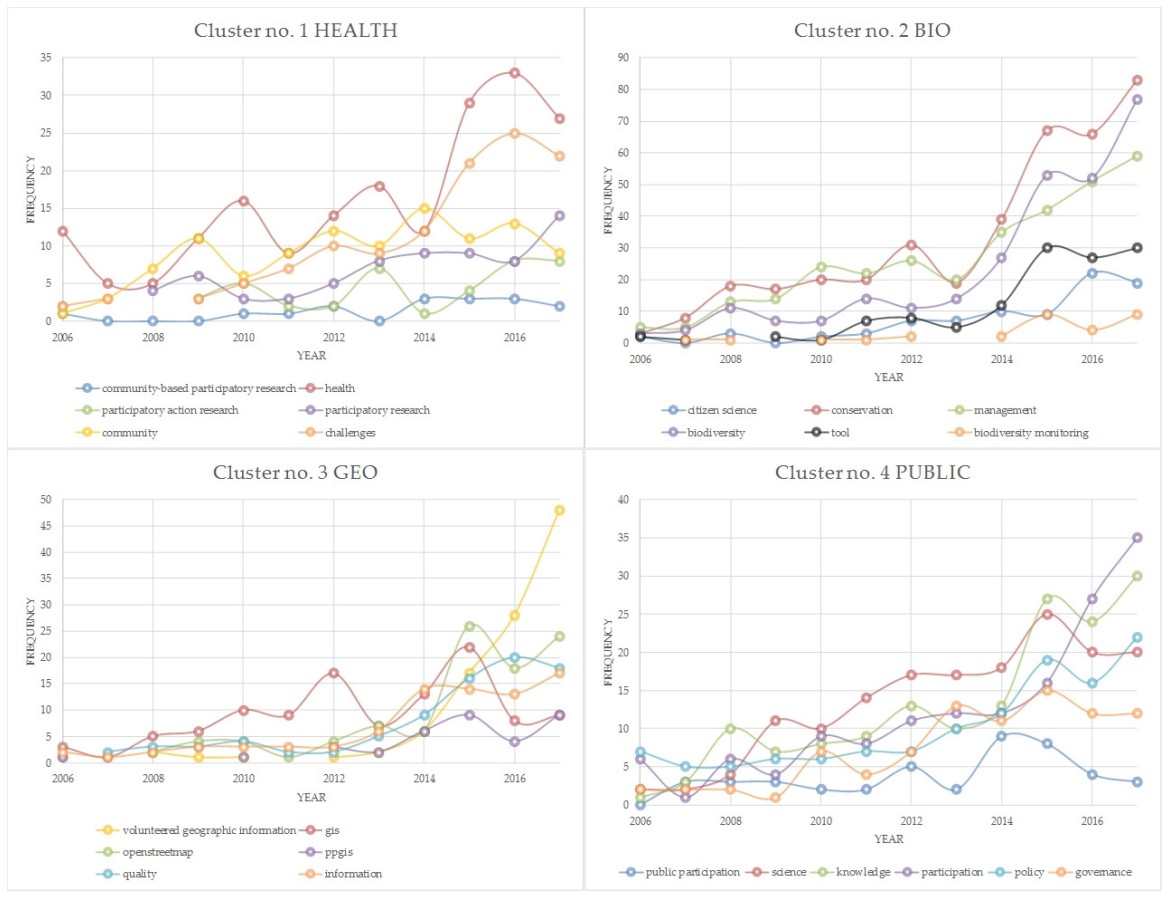

**Figure 7.** Variations in the most frequent terms by cluster.

*3.2. Presence in Social Media*

Of the 3416 documents retrieved, 1895 were mentioned in social media. Inasmuch as the term altmetrics was coined in 2010, only the data for 2012–2017 (which accounted for 86% of such mentions) were included in the analysis discussed below (Table 4). A total of 1629 documents published in those years (66% of the documents with a DOI in the period) were mentioned in social networks. The proportion of documents grew from 56% in 2012 to 72% in 2017.

**Table 4.** Number of documents mentioned in social media.

| Year | Docs w/ DOI | Docs w/ Altmetrics | % | Cumulative % | Docs w/ Altmetrics/Total Docs w/ DOI |
|------|------|------|------|------|------|
| 2006 | 89 | 15 | 0.79 | 0.79 | 16.85 |
| 2007 | 97 | 22 | 1.16 | 1.95 | 22.68 |
| 2008 | 149 | 32 | 1.69 | 3.64 | 21.48 |
| 2009 | 173 | 52 | 2.74 | 6.38 | 30.06 |
| 2010 | 215 | 67 | 3.54 | 9.92 | 31.16 |
| 2011 | 226 | 78 | 4.12 | 14.04 | 34.51 |
| 2012 | 292 | 163 | 8.60 | 22.64 | 55.82 |
| 2013 | 285 | 174 | 9.18 | 31.82 | 61.05 |
| 2014 | 348 | 223 | 11.77 | 43.59 | 64.08 |
| 2015 | 452 | 310 | 16.36 | 59.95 | 68.58 |
| 2016 | 518 | 349 | 18.42 | 78.36 | 67.37 |
| 2017 | 572 | 410 | 21.64 | 100.00 | 71.68 |
| Total | 341 | 1895 | 100.00 | | 55.47 |

The platforms where documents were mentioned most frequently were blogposts and Twitter, followed by Facebook.

In all, 42% of the documents mentioned in social media were openly accessible, three percentage points higher than the mean for open science publications as a whole.

The possible relationship between content and social dissemination was analysed based on the mentions received in each of the four clusters defined. The largest number of documents with mentions was observed for Cluster 2 (BIO) with 872, followed by 1 (HEALTH) with 757, 3 (GEO) with 609 and lastly 4 (PUBLIC) with 516.

As the proportion of open access documents in each cluster and those with mentions in social media graphed in Figure 8 shows, Clusters 1, HEALTH and 2, BIO had the highest percentages of both open access documents and of publications with mentions.

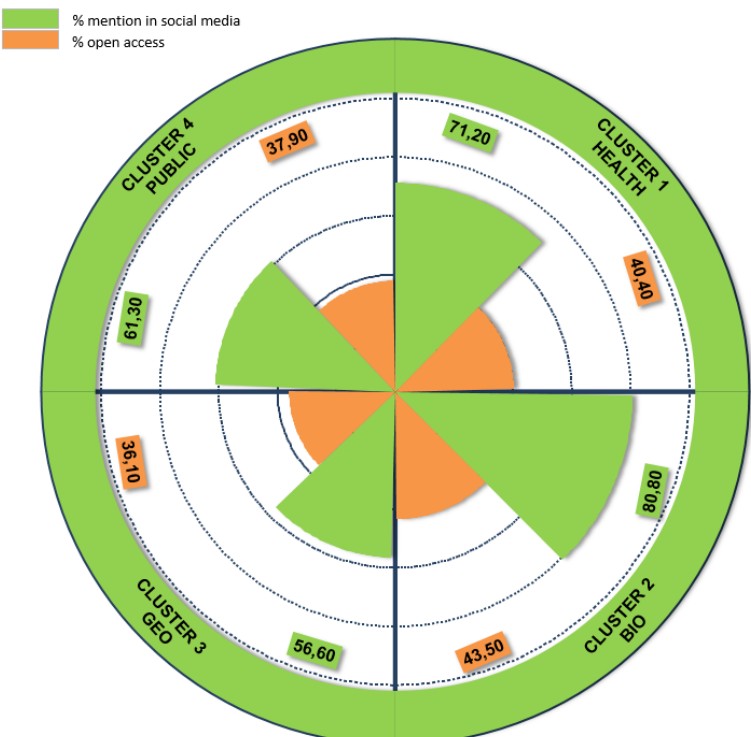

**Figure 8.** Open access documents (%) and documents with mentions in social networks (%).

The relationship between clusters and social networks is shown in Figure 9 (left), where the size of each platform is proportional to the overall number of mentions and the thickness of the lines to the percentage of mentions by cluster. Hence the size of blogposts and Twitter, the two platforms where documents were most frequently mentioned, are similar. Documents classified in Clusters 2 (BIO) and 4 (PUBLIC) had a more significant presence in other networks (such as Facebook and newsfeeds) than the Cluster 3 (GEO) and 4 (HEALTH) publications. Further to the maximum number of mentions (Figure 9, right), Clusters 2 (BIO) and 4 (PUBLIC) were the ones most visible in social networks.

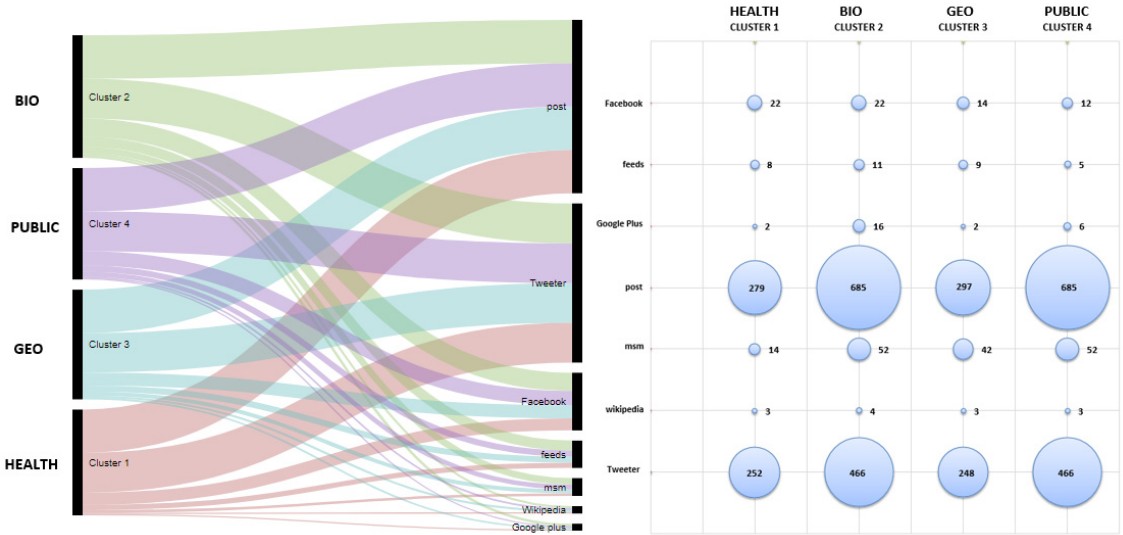

**Figure 9.** Mentions in social networks by cluster (**left**) and maximum number of mentions (**right**).

In all the clusters, but particularly in 4 (PUBLIC) and 2 (BIO), open access documents accumulated the largest number of mentions also in Twitter and in Blogpost (Figure 10).

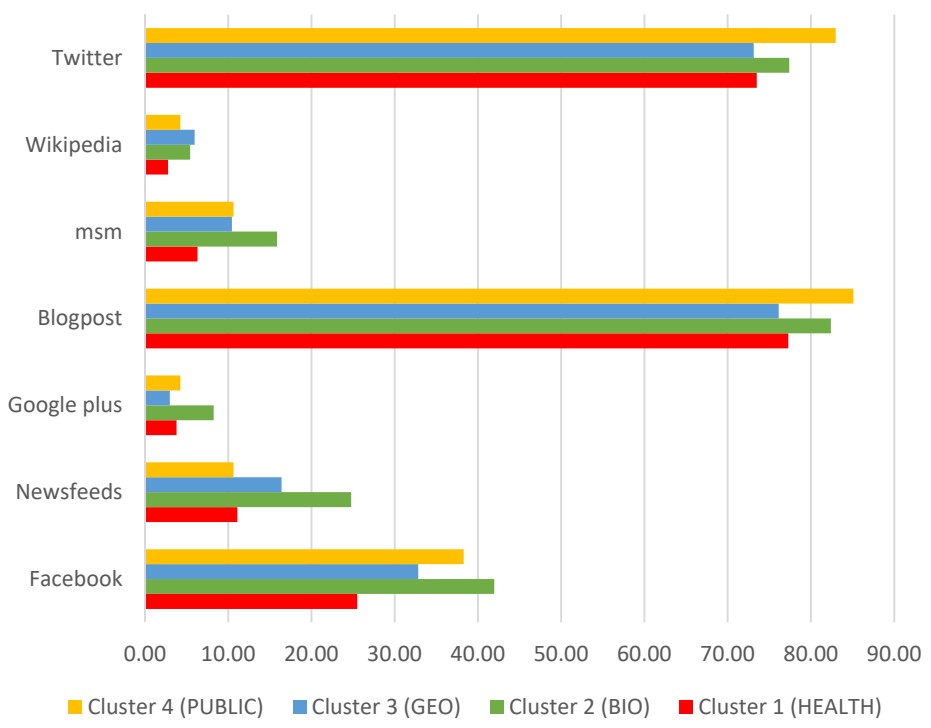

**Figure 10.** Percentage of open access documents with mentions by social network.

## 4. Discussion and Conclusions

Although the change in paradigm in citizen science [26,64] may appear to be a recent development, the subject has a long tradition [25,65] and has given rise to a wealth of scientific publications. The present findings show that interest in citizen science is on the rise, for most of the output was recorded in recent years. Similar studies concur that whilst scientific publications on citizen science were published in the nineteen nineties, growth in the area has been particularly steep since 2010, as mirrored both in the Web of Science [56] and Scopus [57].

This study shows that large output countries such as the US, the UK and European Union Member States in general were also the most productive in this area, although activity was most intense in others, such as South Africa, Canada and Australia. The explanation for such intensity may lie in a series of national and regional strategies implemented in connection with open science policies. Canada has an Action Plan on Open Government 2.0 (OECD, 2015), while South Africa's efforts have been geared primarily to its open access policies, which have earned it [66] acclaim as an example of good open science practice.

A number of European initiatives have been forthcoming to fund citizen science projects. More specifically, ever since the Framework Programmes were instituted in 2002, funds have been earmarked for this endeavour, with scientific publications as one of the most visible academic results. In 2012, for instance, the European Union financed five projects under the topic 'Developing community-based environmental monitoring and information systems using innovative and novel earth observation applications'". The significant concentration in categories such as Environment and Geography in publications on citizen science detected in the present analysis would appear to be related to the implementation of those EU projects.

As reported by many earlier authors, citizen science projects embrace a wide spectrum of areas, including health [67–69], astronomy [28,70] and biology/monitoring biodiversity [27], among others. This study showed that many of those areas are mirrored in the scientific publications identified. Keyword co-occurrence analysis identified four major clusters, HEALTH, BIO, GEO and PUBLIC, which concurred with the categories detected in the literature on citizen science. The initiatives found in HEALTH included studies such as the Mosquito Alert project [68,71]. Examples of BIO-related initiatives were also identified, such as the eBird project [72] and the development of the Global Biodiversity Information Facility (GBIF) [73] database. Studies such as authored by Owen and Parker [74] showed that environmental protection agencies (EPAs) in Europe and the United States are using citizen science ever more frequently for research on the protection of nature. Other authors identified high demand for observation data to inform conservation and environmental policy, reporting that the vast majority of terrestrial biodiversity observations were made by citizen scientists [75], GEO projects were found to include initiatives calling for citizen collaboration using geographic information systems (GIS) to lower the risk of disaster [57]. The subjects found in the PUBLIC or decision-making cluster were observed to be related to urban pollution [24,76] and large city park management [77].

Further to the clusters identified in the two sub-periods studied, research on citizen science has evolved toward new areas. Three clusters were identified in the first period: one focusing on health where traditional methodologies such as community-based participatory (CBPR) and participatory action (PAR) research were observed. The subjects in the BIO cluster were indicative of worldwide concern for conservation of the environment and biodiversity. The PUBLIC-GEO cluster was the only one that included terms associated with information and communication technologies such as "participatory GIS". In the second period GEO was a separate cluster that included geography-related items such as neogeography and OpenStreetMap. Such changes may be a reflection of the digital technology-driven re-vitalisation of citizen science projects [64] that capitalised on the widespread acquisition of easy-to-use smart phones and gadgets to enlist citizen participation in collaborative activities [24].

The traits that differentiate the clusters identified included the purpose sought, the number of participants and the form of their involvement, project scale (local, national, international) and methodology. The success of GEO and BIO projects can be attributed essentially to the vast amounts of data collected and the balance subsequently struck between the quantity and quality of the information processed [78]. As such projects also depend on mass participation, they have benefitted from the popularisation of latest generation mobile devices and ICT tools. These two clusters share methodologies, as revealed by the relationship between them that has given rise to new disciplines such as biogeography [79]. The PUBLIC and HEALTH clusters, in turn, contained terms such as

"engagement" and "partnership", suggestive of the pursuit of long-term commitments on the part of participants, whose active collaboration would be favoured by their simultaneous status as objects of the study.

Another finding of interest was the presence of citizen science papers in the social media. Analysis of that presence, based on the use of altmetric indicators, is highly dependent upon the existence of a DOI. This study showed that the percentage of documents with the identifier grew between 2006–2017, a trend also reported by other authors [80]. The number of documents with mentions in social media grew yearly as well, at rates higher than observed for the percentage of documents with DOI.

This trend was evidenced in previous studies [80,81] in which it is also evident that growth has been especially important in Social Sciences disciplines. In this research, a higher proportion of papers was mentioned in social media (66%) comparing with other studies. In this line, earlier studies showed that the percentage of papers with such mentions ranged from 15% to 24%, with social science and humanities papers ranking highest [41]. Other authors reported values as high as 55% in specific disciplines such as communication. One explanation may be the relationship between the object of study and its social impact, for topics relating to the communication appear to attract substantial attention [81]. This growth of references can be related to an increase and popularization of social media which grew significantly during this period, as well as probably an increased interest of this topic in social media.

The number of tweets is one of the most frequent altmetric indicators, a constant in most studies using Altmetric.com [40,41,49]. Some authors contend that since Twitter is widely used outside academia it may be an especially promising source of evidence for social interest in science [82].

Output on citizen science was observed here to be published under open access arrangements at higher rates than documents on other subjects (30.7% in 2006–2017 compared to the mean 23% for the WoS database as a whole). Such arrangements favour visibility in social networks, for a sizeable percentage of open access documents (42% in 2012–2017) was mentioned in social networks, particularly in tweets. These findings are consistent with results reported by authors such as Bruns and Stieglitz [50].

The relationship drawn between the clusters identified and their visibility in social networks revealed that their profiles were fairly similar, with a predominance of tweets and blogposts in all four, although a larger proportion of BIO and HEALTH documents exhibited a presence in those media. Cluster no.2 BIO and cluster no.4 PUBLIC, in turn, had more mentions in other networks such as Facebook and newsfeeds.

These findings attest to the importance of research on citizen science in recent years and the impact of furtherance policies, due perhaps not only to academic interest, but also to the increasingly clear desire of citizens to play a more active role in knowledge production and decision-making. That notwithstanding, certain shortcomings have yet to be overcome, such as the failure to acknowledge the effort deployed by citizens [75].

Despite the general interest observed, the forthcoming Ninth "Horizon Europa" research and technology framework programme, where open science and open innovation will be two of the three main pillars, does not envisage a programme specifically geared to science with and for society. In light of that absence, organisations such as the Network of National Contact Points for Science with and for Society (Sis.net) have launched petitions to remedy the oversight. Given the indisputable effect that will have on future findings on the subject, monitoring the dynamics of scientific endeavour will play a fundamental role in determining the evolution and social impact of science.

Citizen science, in which the general public becomes involved in research endeavours, furthers their participation not only in terms of effort, but also of tools and resources. The idea is to generate a new scientific culture with improved science–society–policy interaction that will induce more democratic research based on evidence-informed decision-making [26].

**Author Contributions:** Conceptualization, N.B.-P., D.D.F and E.M.; Data curation, N.B.-P., D.D.F and E.M.; Formal analysis, N.B.-P., D.D.F. and E.M.; Funding acquisition, E.S.-C.; Methodology, N.B.-P., D.D.F.; Visualization, N.B.-P. and D.D.F; Writing—original draft, N.B.-P.; Writing—review & editing, N.B.-P., D.D.F., E.M and E.S.-C.

**Funding:** This project received funding from the European Unions' Horizon 2020 Research and Innovation Programme under grant 741657, with the name SciShops.eu. The content of this article does not reflect the official opinion of the European Union. Responsibility for the information and views expressed in the article lies entirely with the authors.

**Acknowledgments:** The authors thank Margaret Clark for her collaboration in translating the text and for her valuable comments.

**Conflicts of Interest:** The authors declare no conflict of interest.

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
