# Peer review of "Scientific Landscape of Citizen Science Publications: Dynamics, Content and Presence in Social Media"

_publications, doi:10.3390/publications7010012_

Reviewer 1 Report

This is a very interesting study and has covered quite a lot of ground with its extensive references and thorough background about open science in general. It is also quite relevant and is of immense value for open science-related researchers. I was impressed by the enormous effort put into the background and literature review, which indicate a comprehensive understanding of the field.

I do have some comments that I believe could enhance this paper enough to be published. But one stands out as an ethical question/concern, with which I would like to start with.

My comment is that the paper does not seem to cover some of the well-known limitations/problems with open science and I was concerned that this may be caused by the fact that the H2020 fund used for it revolves around promoting open science and hence making it difficult for the authors to criticise it objectively. 

To avoid this possible perception, I think the paper could benefit from additional reflections on some of the controversies and criticism surrounding open science (e.g., misuse, verification, quality control, exploitation, ethical issues, etc.) Readers would benefit from learning about the counter argument against open science.

Apart from that, below are some minor comments that I suggest to improve the overall quality, which is very good as it stands:

- On line 123 the authors say "Near universal access to the internet has.." but the reality is that just over 50% of the world population is reported to be online and that is far from universal. A bit of clarity on that would be useful.

- On line 130 the authors refer to the rise to the so-called ‘academic social web’ without further assessment of the challenges this may pose based on the many challenges social media itself have demonstrated through numerous studies. This ties into the above note about looking into some of the flaws that may exist in the practice of open science today.

- On line 149 the authors refer to altmetric indicators but don't explain them sufficiently especially about the 'serious doubts' around them as well as 'certain limitations' (line 203). This would be useful particularly as they use them extensively in the methods.

- The authors use a quantitative approach in identifying articles and mentions using various keywords related to open science. I do not see this as rising to the level of an "analyses" of "the scientific research" itself although it could be seen as an analysis of references and mentions of articles around open science.

- Similarly, on line 179, the authors use the term 'analyse the dynamics' but what the article provides was closer to an analysis of the number of references and citations on social media. 

- On line 197, the authors note "well-known biases" of WoS but do not delve into this despite the fact that it is uses extensively in their research. I recommend some mention of those biases and how they could affect the study.

- On line 214, the authors note that 'Very frequently used terms were included' but methodologically, this is unclear. What is 'very frequently' and where does one draw the line? More effort in explaining this would be useful to assess the methodology.

- On line 241, the authors note that the reliance on a script "developed by the Carlos III University of Madrid’s 241 Information Metrics Studies Laboratory" but no citation or reference to this tool was provided. It would be useful for scientific transparency to understand what open-source libraries were used in the analysis for scrutiny and knowing what possible weaknesses there may have been in the method.

- On 247, the authors refer to the 'percentage of documents with mentions in social media' but without identifying which social media were used. It would have been useful to have that information even as a footnote or link to an online resource of the tool that sheds light on the methodology of data extraction of those social media since many tools have weaknesses.

- On line 288, the authors could have offered a bit of clarification as to what 'green', 'gold' and 'bronze' mean.

- In referring to social media mentions from line 365 onwards, the authors could have mentioned the fact that the growth of references is not only an indicator of more interest but by the mere fact that social media use itself grew tremendously over that period. This brings into question the authors' assumption of "prominence of publications on citizen science in social networks" (line 471 and line 486) because such mentions may have grown regardless if they were on open science or not, also grew significantly. Such data may not exist to draw that conclusion but it is worthy to note that this may not be attributed to a greater interest but purely due to the greater presence of users on social media.

- Finally, the paper would benefit from another proofread as I found a few minor language errors.

Author Response

Find attached the responses. 

Reviewer 2 Report

Authors give proper introduction, explaining terminology and EU science policy. The terms open science and open access are very important for the paper. BOAI and the 2002 definition of OA could be mentioned before the sentence “OA was adopted as a principle in the EU in 2012.” The paper gives a decent overview of literature, including EU regulations, theoretical and research papers concerning open science and citizen science.

The methodology is well explained. However, it is very important to explain all the abbreviations, although some of them are familiar to the potential readers. E.g. it is important to explain what DOI is and why is it important for the research. Also, gold, green and bronze OA should be explained, not only mentioned in quotation marks. Someone who is not familiar with OA will maybe read the paper, not understanding the terms.

The presentation of the results is clear and appropriate. In the Figure 6 word cluster should be written without the accent (the word with the accent is probably Spanish, but the text of the paper should be in English, including figures).

The paper gives a good insight in the state of citizen science and about the productivity of some important countries. Very interesting is the part of the research concerning the presence of citizen science papers in the social media that is analysed based on altmetric indicators. The analysis of the papers in the context of open access is also valuable.

The paper could be interesting for scientists, for wider public and also for  policy makers.

Author Response

Find attached the responses. 
